# Exacerbation of Influenza A Virus Disease Severity by Respiratory Syncytial Virus Co-Infection in a Mouse Model

**DOI:** 10.3390/v13081630

**Published:** 2021-08-18

**Authors:** Junu A. George, Shaikha H. AlShamsi, Maryam H. Alhammadi, Ahmed R. Alsuwaidi

**Affiliations:** 1Department of Pediatrics, College of Medicine and Health Sciences, United Arab Emirates University, Al Ain 17666, United Arab Emirates; junugeorge@uaeu.ac.ae; 2Department of Medical Education, Sheikh Khalifa Medical City, Abu Dhabi Health Services Company (SEHA), Abu Dhabi 51900, United Arab Emirates; shhalshamsi@seha.ae; 3Department of Medical Affairs, Sheikh Shakhbout Medical City, Abu Dhabi Health Services Company (SEHA), Abu Dhabi 11001, United Arab Emirates; mhalhammadi@seha.ae

**Keywords:** IAV, RSV, co-infection, mortality, viral titer

## Abstract

Influenza A virus (IAV) and respiratory syncytial virus (RSV) are leading causes of childhood infections. RSV and influenza are competitive in vitro. In this study, the in vivo effects of RSV and IAV co-infection were investigated. Mice were intranasally inoculated with RSV, with IAV, or with both viruses (RSV+IAV and IAV+RSV) administered sequentially, 24 h apart. On days 3 and 7 post-infection, lung tissues were processed for viral loads and immune cell populations. Lung functions were also evaluated. Mortality was observed only in the IAV+RSV group (50% of mice did not survive beyond 7 days). On day 3, the viral loads in single-infected and co-infected mice were not significantly different. However, on day 7, the IAV titer was much higher in the IAV+RSV group, and the RSV viral load was reduced. CD4 T cells were reduced in all groups on day 7 except in single-infected mice. CD8 T cells were higher in all experimental groups except the RSV-alone group. Increased airway resistance and reduced thoracic compliance were demonstrated in both co-infected groups. This model indicates that, among all the infection types we studied, infection with IAV followed by RSV is associated with the highest IAV viral loads and the most morbidity and mortality.

## 1. Introduction

Respiratory infections are major causes of both morbidity and mortality worldwide [1]. Diagnostic and surveillance studies have revealed that co-infections with more than one pathogen in the respiratory tract are common [2,3]. Pathogens involved in co-infections interact with the host and with themselves, thereby profoundly affecting replication of each pathogen, disease pathogenesis, immune responses and most notably the disease outcome [4,5].

Influenza A virus (IAV) and respiratory syncytial virus (RSV) are common causative agents of respiratory tract infections among all age groups worldwide. RSV is the most common cause of lower respiratory tract infections in children. IAV causes seasonal influenza that affects 5–10% of adults and 20–30% of children every year [6,7,8]. IAV is a negative strand, enveloped RNA virus that infects airway epithelial cells [9], whereas RSV is an enveloped virus that contains a linear negative-sense RNA genome. Simultaneous co-infections with these two viruses are frequently associated with lower respiratory tract illness in infants [10]. Pediatricians are often challenged when it comes to discriminating between the two viruses, as they have similar clinical presentations. RSV and influenza might have a common ecologic niche in the respiratory tract. In vitro experiments suggest that while RSV grows in MDCK cells, co-infection with IAV leads to a reduction of progeny RSV [11]. Consistently, epidemiologic studies show that when RSV infections are widespread, influenza infections are scarce; the opposite is also true [12]. When individual cells are co-infected, one virus strain usually influences the replication of the other virus, a phenomenon termed viral interference. The result is often the exclusion of one virus but persistence of the other [13]. Better understanding of co-infection with both viruses regarding disease severity is required for optimal patient care and the development of effective preventive and therapeutic strategies. In this study, a mouse model was established based on co-infection with influenza A (H1N1 A/PR/8/34) and RSV A2 with the aim of revealing the in vivo effects of RSV and IAV co-infection on disease severity, as reflected in morbidity, survival, viral load, immune responses and lung function.

## 2. Materials and Methods

### 2.1. Materials and Reagents

Trizol (Invitrogen, Carlsbad, CA, USA); MMLV reverse transcriptase kit (Promega, Madison, WI, USA). Universal PCR Master Mix, primers and probe were all obtained from Applied Biosystems™ (Foster City, CA, USA). Fluorochrome-conjugated antibodies (CD4-APC, CD8-APC-Cy7, CD19-PE-Cy7, CD11B-PECY7) used for immunophenotyping analysis were produced by Bio Legend (San Diego, CA, USA). Ketamine (50 mg/mL), xylazine (methanol at 10 mg/mL), phosphate-buffered saline (PBS), Trypan blue, RPMI-1640, NH_4_Cl, KHCO_3_ and EDTA were purchased from Sigma-Aldrich (St. Louis, MO, USA). Filters (40 and 100 µm) were a product of Corning (Corning, NY, USA).

### 2.2. Animals and Virus Inoculation

BALB/c mice (20–22 g), obtained from Jackson Laboratory (Bar Harbor, ME), were housed at 22–25 °C with a 12-h light/dark cycle in sterile cages and fed ad libitum with standard rodent chow and filtered water. Mice were acclimatized for a week before they were inoculated intranasally with a dose of 2 × 10^6^ TCID_50_/uL, 10 uL per nostril of IAV H1N1 A/PR/8/34, and/or RSV A2, 10^7^ PFU/mL, 50 uL per nostril, under ketamine–xylazine sedation as described previously [14].

### 2.3. Experimental Strategy

An equal number of age and sex matched BALB/c mice were divided into five groups. A group of mice was intranasally inoculated with PBS and constituted the uninfected control group, and two other groups served as IAV and RSV single-infection groups. The remaining two experimental groups were challenged with both viruses, which were administered sequentially, 24 h apart. Accordingly, animals in the fourth group were inoculated first with IAV and then challenged with RSV 24 h post IAV infection (IAV+RSV). Mice in the fifth group were pre-challenged with RSV 24 h prior to IAV inoculation (RSV+IAV). The day of inoculating mice with the single viral infection was considered day 0. The day of inoculating mice with the second virus in co-infected groups was considered to be day 1, (Figure 1). Mice in all 5 groups were monitored daily for changes in body weight and death for up to 7 days.

### 2.4. Sample Processing

On days 3 and 7 post-infection, mice were euthanized using urethane (25% *w/v*), as described previously [15]. After euthanasia, whole lungs, thymus and spleen samples were excised, weighed and analyzed immediately. Lungs samples were processed for viral gene quantification by real time PCR (qPCR), and lung tissues were processed for immunophenotyping by flow cytometry.

### 2.5. Quantitative PCR for Determining Viral Loads

Expression levels of the RSV nucleocapsid (N) gene, IAV PR8 matrix gene and beta actin (β-ACTIN) in control and infected mice were assessed using a two-step real time PCR protocol involving RNA extraction, cDNA synthesis and real time PCR amplification, as described previously [14,16]. Briefly, total RNA was extracted from lung tissue using trizol and then reverse transcribed using the MMLV reverse transcriptase kit. RNA was quantified by using NanoDrop Spectrophotometer (Thermo Scientific™, Waltham, MA, USA). Real-time PCR was performed with the primers described in Table 1, using TaqMan^®^ Universal PCR Master Mix (Applied Biosystems™, Waltham, MA, USA). The amplification conditions included an initial denaturation step at 94 °C for 5 min, followed by 40 cycles of denaturation at 94 °C for 45 s and annealing at 60 °C for 1 min. A standard curve was prepared using cDNA prepared from virus samples of known titer; IAV = 10^8.1^ TCID_50_; RSV = 3.7 × 10^7^ pfu/mL. Viral titer in the lung samples was calculated by extrapolating from the standard curve. The viral load per lung was calculated based on the weight of lung.

### 2.6. Flow Cytometry by Fluorescence Activated Cell Sorting (FACS)

Lung tissues from mice were minced into small pieces and incubated in RPMI-1640 containing 0.2 mg/mL collagenase type IV (Sigma–Aldrich) for 45 min at 37 °C. DNase (10 U/mL) was added to the digested lungs. Digested lungs were filtered using 100 um and then through 40 um filters. The pellet was washed two more times and re-suspended in 1 mL erythrocyte lysis buffer for 10 min, after which the lysis buffer was washed off. The erythrocyte-free pellets were re-suspended in 1 mL of FACS buffer. An aliquot of cells was stained with trypan blue to determine the cell number and viability. Then, 10^6^ cells were incubated at 4 °C for 30 min in the FACS buffer (PBS with 1% FBS, 0.1% sodium azide) containing Fc receptor blocking antibody (CD16/32/clone 93). After washing, cells were stained for 30 min with 7AAD dye along with fluorochrome-conjugated antibodies (CD4-APC, CD8-APC-Cy7, CD19-PE and CD11B-PECY7). The cells were washed and re-suspended in 300 µL PBS. A minimum of 50,000 events were acquired and analyzed on BD FACSCanto II, using BD FACS DIVA software (BD Bioscience, San Jose, CA, USA). Non-viable cells with positive 7AAD staining were excluded from analysis. Gating of surface markers or compensation was determined using control samples and the unstained approaches, as described previously [17]. Briefly, total live cells were identified based on the exclusion of a 7AAD dye. Thereafter, live cells were gated as CD4/CD8 (T cells) in one dot plot, CD19 (B cells) and CD11B (macrophages) in another dot plot.

### 2.7. Lung Function Assessment

We utilized the flexiVent instrument from SCIREQ (Montreal, PQ, Canada) to perform pulmonary function analysis, as described previously [14]. Briefly, on day 3 post infection, lung function parameters, including thoracic resistance (Rrs, cmH_2_O.s/mL), thoracic compliance (Crs, mL/cmH_2_O) and large airway resistance (Rn, cmH_2_O.s/mL), at baseline and after methacholine challenges were measured in tracheotomized mice.

### 2.8. Statistical Analysis

Experimental data are expressed as mean ± standard error of the mean (SEM). Comparison among groups was performed using the nonparametric test (2 independent variables: Mann–Whitney) and *p* value < 0.05 defined statistical significance.

## 3. Results

### 3.1. Body Weight Loss Was More Severe in Mice Infected with IAV 24 h Prior to RSV Inoculation

To evaluate the effects of co-infection on disease severity, mice were monitored daily for mortality, (Figure 2A) and body weight loss (Figure 2B) over 7 days. Co-infected mice lost weight between days 2 to 7 at a higher rate than mice infected by IAV or RSV alone. Mice inoculated with RSV alone did not exhibit any weight loss. Mice infected with IAV alone or in combination showed severe weight loss. The weight loss was more pronounced in mice that received IAV followed by RSV (IAV+RSV). Consistently, mice in this group had 50% mortality by day 6. The mice in the other groups all survived until the end of the study.

### 3.2. RSV Infection following IAV Results in Remarkable Increase in IAV Viral Load in the Lungs on Day 7 Post-Infection

We next examined whether the significant weight loss and death observed with co-infection was due to increased IAV replication in the lungs. On day 3, IAV viral load was highest in IAV+RSV group compared to other IAV infected mice but did not reach statistical significance, (Figure 3A). However, on day 7, the viral load was one log higher in IAV+RSV group than other groups. For example, there was a significant reduction in the viral load in groups receiving IAV alone in comparison to IAV+RSV group.

### 3.3. IAV Infected Mice Exhibit Reduced RSV Viral Load following Subsequent RSV Infection on Day 7 Post-Infection

The RSV viral load was increased on day 3 in RSV, RSV+ IAV and IAV+RSV groups, (Figure 3B). However, the co-infection groups showed different kinetics on day 7. The RSV viral load was significantly reduced in IAV+RSV group by day 7. A similar trend was observed in RSV+IAV as well but did not reach statistical significance. On the contrary, the group receiving RSV prior to IAV infection was able to show RSV infection, and the viral titers showed a decreasing trend by day 7 but less so than those observed in the RSV-alone group.

### 3.4. Co-Infection of IAV with RSV Leads to Significant Weight Loss of Immune System Organs and Significant Enlargement of Lungs

In order to observe the effect of viral replication on the immune system, the weights of the lungs, spleens and thymuses of mice euthanized on days 3 and 7 post infection were determined. As expected, on day 3, the weight of a lung in each group, the site of viral replication, (Figure 4A), was far higher than in the uninfected control group, except the group receiving RSV alone. On day 7, the weight of the lung was much higher in co-infected groups than singly-infected and uninfected groups. Commensurate with the observed viral load, the IAV+RSV group showed the steepest increase in lung weight: it was almost three times that of control mice. The IAV alone and RSV+IAV groups also showed significantly higher lung weights than control mice, albeit lower weights than the IAV+RSV group. Surprisingly, the group infected with RSV alone did not show any increase in lung weight.

The changes in weight of lymphoid organs such as the spleen (Figure 4B,E) and thymus (Figure 4C,F) were also evaluated. All infected groups showed reduced spleen weight compared to the control group. The least reduction was observed in the RSV-alone group. On day 7, weight loss was less pronounced than on day 3. The maximum reductions were observed in IAV+RSV and RSV+IAV groups at both time points. Thymus weight was considerably reduced in IAV alone, IAV+RSV and RSV+IAV groups. However, the RSV-alone group did not show any reduction in thymus weight.

### 3.5. Co-Infection Resulted in a High Number of CD8 T Cells and Low Numbers of CD4 T Cells and B Cells

We investigated the relative abundances of key immune cells, including CD4 T cells, CD8 T cells and B cell lymphocytes in the lungs of mice on day 7 post infection. There was a marked reduction in the proportion of CD4 T cells in every group on day 7, except in mice infected with RSV. The IAV-alone group showed a significant increase in the CD4 T cell population when compared to control mice (Figure 5A). The CD8 T cell population was considerably elevated in all experimental groups except the RSV-alone group (Figure 5B). The maximum increases were observed in co-infection groups. Both had roughly a 5-fold increase over the control and 1.5-fold more than in the IAV-alone group. The RSV group had the lowest count of cytotoxic T cells on day 7. On the contrary, the B cell population, identified as CD19 cells (Figure 5C), showed a reducing trend in co-infected groups compared to singly-infected groups, but the differences were not statistically significant. Macrophages (CD11B positive population) (Figure 5D) showed a marked increase by day 7 in every group except the RSV-alone group, compared to the control group.

### 3.6. Co-Infection with IAV+RSV or RSV+IAV Leads to Increased Thoracic Resistance and Large Airway Resistance

In order to determine the consequence of co-infection on pulmonary function, we measured lung function changes on day 3 post infection. In co-infected groups, lung function impairments were severe compared to control mice. Said groups mainly showed increased thoracic and large airway resistances and decreased thoracic compliance (Figure 6). The data demonstrate significant, almost 2-fold increases in thoracic resistance and central airway resistance in co-infected mice. 

## 4. Discussion

In this study, a murine model of IAV and RSV co-infection was utilized to evaluate disease severity markers among single IAV and RSV infections in comparison to co-infections with both viruses. Our results indicate that RSV inoculation 24 h before IAV infection results in higher IAV viral loads than IAV infection alone on day 7 post-infection and leads to severe morbidity and mortality. In contrast, RSV viral load was reduced in the IAV+RSV group by day 7 in comparison to single RSV infection. Co-infection with IAV and RSV was also found to result in increased CD8 T cell recruitment in the lungs. Additionally, lung function impairment was more pronounced in groups co-infected with both viruses.

Gonzalez and colleagues have shown that infection of mice with rhinovirus 2 days before IAV infection reduced the disease severity when using a low or medium, but not a high dose of IAV (PR8 strain) [18]. In another study, concurrent infection of BALB/c mice with RSV and IAV (Pr/H3N2 strain) was associated with reduced disease severity and the downregulation of viral virulence markers such as IAV PB2, PB1, M1, RSV F and M genes. Remarkably, single and priori infections (i.e., inoculation 24 h earlier) in the same study resulted in consistent upregulation of the viral virulence markers [19]. In our study, we demonstrated that IAV (PR8 strain) and RSV (A2 strain) co-infections always resulted in severe disease (increased morbidity (weight loss) and mortality (Figure 2)) compared to single infections. The differences in disease severity between our study and earlier studies are possibly related to using different IAV strains (PR8 vs. Pr/H3N2), variable IAV doses (low vs. high), the timing of co-infection (24 h earlier vs. concurrent) and co-infection with another virus (rhinovirus vs. RSV). 

In this study, when IAV infected mice received a subsequent infection with RSV, increased IAV viral load in lung was evident on day 7 post-infection in comparison to single IAV infection (Figure 3A). Similar results were also observed in a IAV+rhinovirus co-infection model [18]. Our findings suggest that subsequent RSV infection does not directly inhibit IAV replication and probably leads to delayed IAV viral clearance from the lungs of mice in the IAV+RSV group compared to other groups. In vitro studies have also shown that RSV fails to inhibit the replication of IAV [11]. In contrast, parainfluenza virus has been shown to enhance IAV replication [20].

RSV is a relatively slowly replicating virus and often has a lingering course of disease. It is worth noting that RSV is a strong inducer of IFN in most cells [21]. Our study showed that the RSV viral load was significantly reduced in the IAV+RSV group by day 7 (Figure 3B). In a ferret model, infection with the 2009 pandemic’s influenza A H1N1 virus prevented subsequent infection with RSV [22]. Consistently, infection with IAV was found to inhibit the ability of RSV to infect mouse lung cells in association with the upregulation of interferon-induced protein with the tetratricopeptides [23]. Therefore, earlier IAV infection might have neutralizing immunopathological effects in the respiratory tract, leading to the observed viral interference phenomenon.

Although both RSV and IAV induce lung pathology, the resulting pathological consequences are distinct. IAV infection is known to cause severe lung injury characterized by increased alveolar epithelial permeability, leakage of protein containing fluid into the alveolar space and infiltration of immune cells into the lungs, resulting in increased lung weight associated with poor gas exchange and ultimately respiratory failure [24]. In our study, the increase in lung weight showed divergence in co-infections. IAV+RSV co-infection led to the greatest increase in lung weight, but RSV+IAV did not show much difference in lung weight in comparison to the single IAV infection group (Figure 4A,D), supporting a proposed protective role for RSV against subsequent IAV infections. Chan et al., in their ferret model, demonstrated that prior infection with RSV significantly reduced morbidity, as measured by weight loss, after challenge with IAV (H1N1) but did not prevent infection, and all animals were shedding both viruses [22].

Viruses have good immunosuppressive machinery to overcome host immune responses. The apoptosis of immune cells is a common feature of most viral infections. The apoptosis may be induced by direct viral infections of immune cells, or a cytokine storm induced by viral infections elsewhere in the body. The changes in immune organ size provide a broad idea on the effect of a virus on the immune system. IAV infections, either singly or as co-infections, invariably led to reductions in spleen size (Figure 4B,E). However, in RSV-infected mice, the reduction in spleen size was less profound. The thymus is the primary lymphoid organ of the body. IAV infections in mice are characterized by drastic loss of thymus weight. The reduction in thymus weight was also evident even in co-infection conditions. However, the mice infected with RSV alone did not show any significant change in thymus weight (Figure 4C,F).

Lymphocytes play a crucial role in the clearance of a virus from a host. They also contribute to immune response-induced tissue damage. As reported earlier, IAV and RSV induce strong T cell responses in the lungs of infected animals [25]. The observed reduction of CD4 T cells in co-infected mice is very intriguing (Figure 5A). CD4 T cells are responsible for the cytokine response, and the reduced immunopathology reported in a rhinovirus and influenza co-infection model may correlate well with a reduced CD4 T cell count [18]. However, in that model, rhinovirus infection led to attenuation of disease in IAV infected mice. 

Antigen presenting cells and CD8 T lymphocytes have been associated with increased severity of symptoms, although this has been shown to enhance viral clearance [26,27,28]. CD8 T cells are supposed to be the primary weapon for clearing virus infected cells from the host. Any reduction in the number or functionality of these cells can result in persistence of virus-producing cells for a long time and an increase in viral load in the lung. The CD8 T cell abundances in the cohorts co-infected with IAV were much larger than in single infection groups. Even in high numbers, these cells failed to control the IAV viral titer in infected mice. Further studies are needed to ascertain the functional fitness of these CD8 T cells. The RSV control was found to be better in the co-infected group than in the single infection group, which might have been due to inhibition or competition by IAV.

Excess mucus production is a common feature of respiratory viral infections and is associated with mechanical airway obstruction resulting in impairment of airway resistances [29]. Airway resistance is the change in transpulmonary pressure needed to produce a unit flow of gas through the airways of the lung [30]. In this study, we utilized a forced oscillation system to measure thoracic resistance (Rrs), thoracic compliance (Crs) and large airway resistance (Rn) at baseline and after methacholine challenge. On day 3 post-infection, we observed increased thoracic and large airway resistances and decreased thoracic compliance in co-infected groups in comparison to singly-infected groups (Figure 6), coinciding with the elevated viral loads and increased morbidity and mortality described earlier. Lung function on day 7 post-infection could not be evaluated due to mice mortality in co-infected groups.

Our study has limitations. BALB/c mice are described as semi-permissive hosts of human RSV (RSV A2), which may have contributed to the differences in disease severity observed in our experiments [31]. Although ferrets are a better model for understanding the pathology of influenza viruses in humans [32], BALB/c mice are easily accessible, and we had prior experience using them to study IAV and RSV infections [14,15,33,34]. As we used a mouse adapted IAV strain (PR8), further studies using circulating clinical isolates of both RSV and IAV are needed to provide more realistic data. Furthermore, future studies are also needed to explore IAV and RSV co-infections using different timing for viral inoculations (e.g., simultaneous inoculation with both viruses, 48 h later, 36 h later, etc.) and to study the effects of using different doses (viral inoculum) of both viruses.

In conclusion, this study showed that infection with IAV followed by RSV is associated with higher IAV viral loads in comparison to IAV infection alone on day 7 post-infection. Morbidity and mortality are also more likely. In contrast, the RSV viral load was reduced in mice previously infected with IAV. The findings of this study revealed disease markers that are important for a better understanding of the viral interactions during co-infections with IAV and RSV. This knowledge may facilitate the development of novel strategies to prevent or ameliorate respiratory viral infections.

## Figures and Tables

**Figure 1 viruses-13-01630-f001:**
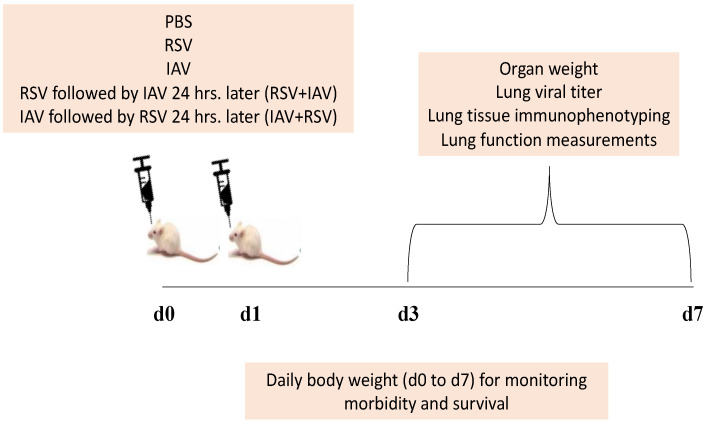
A schematic representation of the experimental strategy for the RSV and IAV co-infection model. Mice were intranasally infected with RSV A2, IAV PR8 or both, one after the other, after a 24 h interval (RSV+IAV or IAV+RSV). One group of mice was intranasally inoculated with PBS and served as the control group. All groups were monitored for up to 7 days (daily weight and survival); on selected days post-infection (d3 and d7), mice were sacrificed for experimental analyses that included organ (lung, spleen and thymus) weights, viral titers, immunophenotyping and lung function.

**Figure 2 viruses-13-01630-f002:**
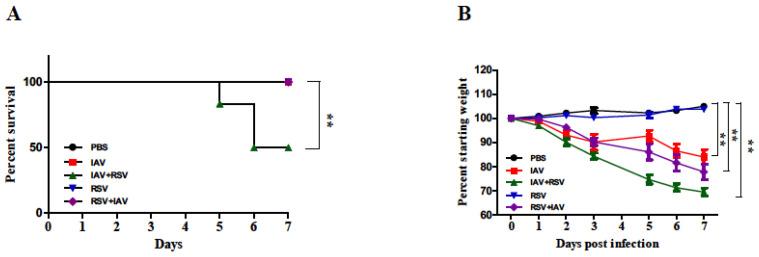
Body weight loss was more severe in mice infected with IAV 24 h prior to RSV inoculation. Mice were infected with IAV PR8, RSV A2 or one after the other, 24 h apart (IAV+RSV and RSV+IAV) and monitored for (**A**) survival and (**B**) body weight changes for 7 days. Mice inoculated with PBS served as the control. Values are mean ± SEM (in percentages) of the daily weight divided by the starting weight for each mouse. Survival data were derived from two independent experiments (*n* = 6 mice per group). Body weight data were derived from five independent experiments (*n* = 18 mice per group). Asterisks denote significant differences between the control and experimental groups; (** *p* < 0.005). Survival analysis was performed by Kaplan–Meier survival curves and a log-rank test, using GraphPad Prism.

**Figure 3 viruses-13-01630-f003:**
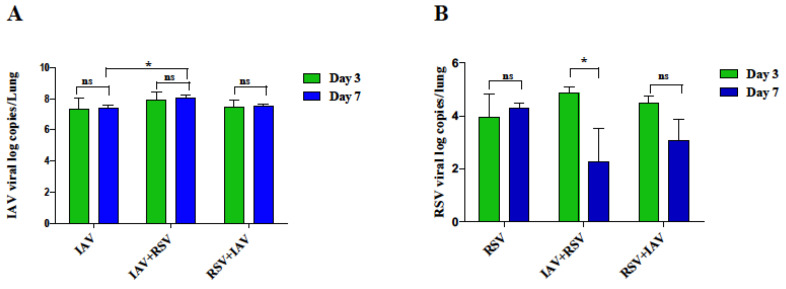
IAV and RSV lung viral loads in co-infection mouse model. On day 3 or 7 post-infection, mRNA was extracted from lungs and processed for real time PCR analysis of (**A**) IAV and (**B**) RSV genes. Results represent mean pooled values +/− SEM from three independent experiments, (*n* = 5–9 mice per group). Asterisks denote significant differences between experimental groups; * *p* < 0.05; ns = not significant. Comparison among groups was performed using the nonparametric test (2 independent variables: Mann–Whitney).

**Figure 4 viruses-13-01630-f004:**
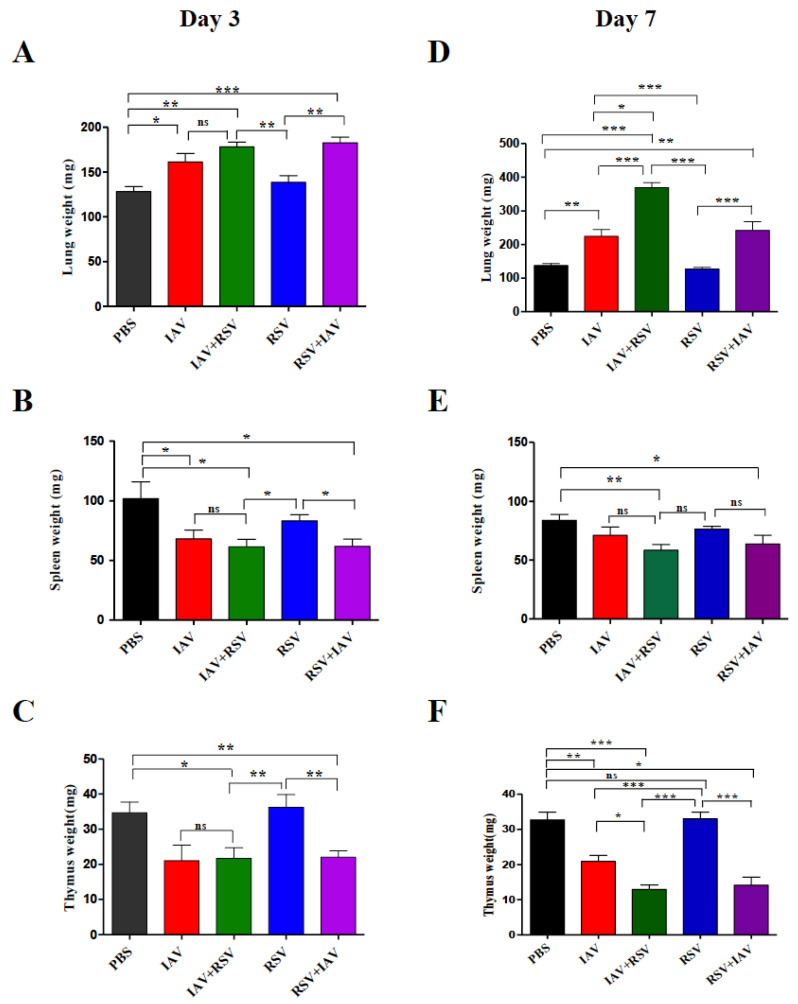
Co-infection leads to significant atrophy of the spleen and thymus, and an increase in lung weight. On day 3 or 7 post-infection, mice were sacrificed, and lung (**A**,**D**), spleen (**B**,**E**) and thymus (**C**,**F**) were excised and weighed. Results represent mean pooled values +/− SEM from three independent experiments. Asterisks denote significant differences among the control, co-infected and singly-infected groups; (* *p* < 0.05, ** *p* < 0.005, *** *p* < 0.0005); ns = not significant. Comparison among groups was performed using the nonparametric test (two independent variables: Mann–Whitney).

**Figure 5 viruses-13-01630-f005:**
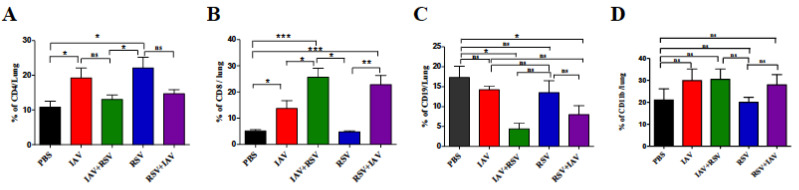
The effects of co-infection on immune-cell populations in the lungs. On day 7 post-infection, mice were sacrificed, and lung immune cells were stained with anti-CD4 (**A**), anti-CD8 (**B**), anti-CD19 (**C**) and anti-CD11B (**D**) antibodies and analyzed by flow cytometry. Results represent mean pooled values +/− SEM from two independent experiments. Asterisks denote significant differences among the control, co-infected and singly-infected groups (* *p* < 0.05, ** *p* < 0.005, *** *p* < 0.0005); ns = not significant. Comparison among groups was performed using the nonparametric test (two independent variables: Mann–Whitney).

**Figure 6 viruses-13-01630-f006:**
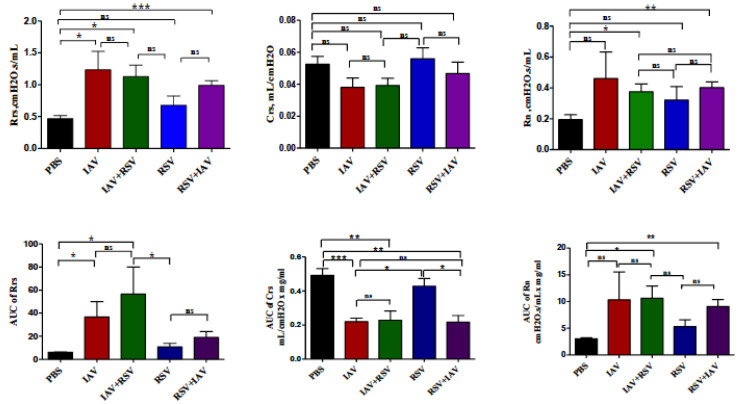
Lung function assessment on day 3 post-infection. Upper panels: Baseline airway reactivity expressed as thoracic resistance (Rrs), thoracic compliance (Crs) and large airway resistance (Rn). Lower panels: Methacholine responsiveness shown as “area under the curve” (AUC) of Rrs, Crs and Rn against methacholine concentration. The values are mean ± SEM. Two separate experiments (*n* = 6) were performed. Asterisks denote statistically significant differences between experimental groups and the control group (* *p* < 0.05, ** *p*  <  0.005, *** *p* < 0.0005). Two separate experiments (*n* = 6) were performed; ns = not significant. Comparison among groups was performed using the nonparametric test (two independent variables: Mann–Whitney).

**Table 1 viruses-13-01630-t001:** IAV and RSV primers that were used in the study.

Primer	Sequence
RSV N Forward	5′GCTCTTAGCAAAGTCAAGTTGAATGA3′
RSV N Reverse	5′TGCTCCGTTGGATGGTGTATT-3′
RSV N Probe	FAM/ACACTCAACAAAGATCAACTTCTGTCATCCAGC TAMRA
IAV Forward	AGA TGA GTC TTC TAA CCG AGG TCG
IAV Reverse	TGC AAA AAC ATC TTC AAG TCT CTG
IAV Probe	FAM-TCA GGC CCC CTC AAA GCC GA-TAMRA

## Data Availability

The data presented in this study are available on request from the corresponding author.

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
