# Peer review of "Exacerbation of Influenza A Virus Disease Severity by Respiratory Syncytial Virus Co-Infection in a Mouse Model"

_viruses, 2021, doi:10.3390/v13081630_

Round 1
Reviewer 1 Report
In this study, George et al investigated the effect of IAV and RSV infections individually, or co-infections on mice. They assessed the morbidity (weight loss) and mortality of these mice on days 3 and 7 post infection. They also assessed lung function, viral titre, organ weight and cellular infiltrates in these different groups. They found that RSV alone had limited effects on mice. Mice infected with IAV were generally more effected, however co-infection with IAV followed by RSV had the greatest effect on mice. This is an elegant and well executed study and I have several very minor comments to the authors.
Introduction
The introduction was concise and explained all of the relevant background to understand the study
Methods
The methods were very well written and easy to follow. In particular the 5 groups of mice were easy to understand.
The authors should amend the sentence about trypan blue staining (line 116-117), as it is currently written, it appears all cells were stained with trypan blue before antibody staining (rather than just a subset for counting purposes).
Could the authors please include which BD Flow cytometer was used as the current “BD FACS DIVA” (line 122) refers to the software and not the actual instrument.
The authors should include a simple sentence on the gating strategy used so that readers do not have to refer to another paper.
Results
The results were generally well written and easy to follow.
There are a few interpretive sentences throughout the results sections “this suggests that mice inoculated with XXX” (line 172 and others) which are not needed as results should just describe the findings, and would be better left for the discussion.
It would be great if the authors could include “ns” on any of the figures where stats have been done so the reader knows what has been compared but is not significant vs what hasn’t been compared.
I am not sure the comment “RSV replication was very slow in the infected mice” (line 171) is useful or that there is any evidence of this.
There are so me formatting inconsistencies in the figures, particularly within the axis labels of the graphs.
I would consider rearranging the order of graphs in Figure 4 to fit with the flow of the text. Ie lungs first, followed by spleen and thymus. I would consider making lung D3 A, and lung D7 B etc to make it easier to follow and easier for the figure legend.
The authors should indicate that B cells are being measured by CD19 staining (line 217) and elaborate on what “showed a reducing trend” (line 218) means.
The authors should indicate what CD11b is being used to measure in the text.
The figure legends are well written and informative. Could the authors please include what statistical tests have been done? I see they are mentioned in the methods but would be good to name them in the legends also.
The figures would benefit from some raw data, some FACS plots (Fig 5) and perhaps photos of mice or organs etc for other figures if the authors have any.
Discussion
I found the discussion a little bit hard to follow and found that ideas were repeated. The flow and ease of reading would be greatly enhanced if the discussion was slightly re-structred and ideas were grouped and discussed before moving on. For example, discussing how the co-infection could be working including all evidence from other models etc.
“RSV load was significantly reduced” (line 271) this was not shown with stats in the figure
There are several instances where the authors have referred to previous studies without referencing relevant literature.
It would be useful if the authors could elaborate on the limitations and future directions of their findings.
Could the authors comment on their choice of mouse model? Mice are certainly more accessible than ferrets for example, however ferrets are a better model for understanding the pathology of influenza virus in humans. Perhaps the authors could comment within the discussion about how they would expect their results to translate into humans and how/what additional studies could be completed to translate their findings (related to the above comment re limitations and future directions).
The authors should be careful not to overinterpret their findings, particularly in the results section (hence why removing any interpretation as mentioned above is advised). Just because a viral titre is higher or lower, doesn’t necessarily mean that the effect is promoting infection/replication etc. For example, the authors suggest that RSV infection after initial IAV infection enhances viral replication because the titres are higher. Instead it may prevent viral clearance? These ideas/theories can, and should be discussed, with references to other literature but should not be the definitive conclusion as this was not shown in this study. As such, I do not agree with the definitive nature of the concluding sentence (that it promotes replication) and should be toned down a bit. For example, it would be accurate to say that “subsequent increases viral load over IAV infection alone at D7, and leads to severe morbidity and mortality” however the mechanism behind this (promoting replication) has not been definitely proven in this manuscript.
Reviewer 2 Report
Line 246-247 - In contrary - should read In contrast or Contrary
Line 265 RSV infection do not directly should be does not directly
Line 298 - should read " the lungs of infected animals"
Line 318 should not be capitalized - Infection
You indicated that you used 10-18 mice per group. Why is there such a disparate number - nearly double in some groups. I ask this as you present your results in % survival which could be significant if the IAV-RSV group had many less animals than other groups.
Reviewer 3 Report
The study describes an interesting experiment based on a clever conceptualisation and idea regarding the co-infection of mice with Influenza A virus (IAV) and respiratory syncytial virus (RSV). The manuscript is in general well written and well presented, whereas the experimental procedure is generally correct. There are some mistakes that need to be corrected stated below:
line 35 and 36: please replace “causes” with “causative agents”
line 36-37: Do you mean worldwide or in a particular country? Please clarify
lines 47-48: It is more appropriate to say “one virus molecule” or “one virus strain” than “one virus”
lines 99-109: Please provide some more details, i.e. the primers used, the conditions etc.
line 265: please correct and replace “do” with “does”
The main problem that has also to be corrected in the description of the molecular procedures in Materials and Methods section. Particularly, more details have to be provided, i.e. the primers used, the conditions etc. Also, in line 105 the "TaqMan® Universal PCR Master Mix and Applied Biosystems™" has to be replaced with "TaqMan® Universal PCR Master Mix (Applied Biosystems™)".
Also details for the method of quantification have to be described. How did the authors use the Ct value? How was the standard curve prepared?
How did the authors use the calculated copy numbers to compute the quantification?
I also recommend to explain better, probably in the introduction, the term "airway resistance".
Finally, it would be valuable to explain if apart from weight loss the researchers did also observe reduced food intake. It is important to discuss this finding if it is so
Round 2
Reviewer 3 Report
The manuscript has been improved and therefore I suggest publication in its current form
Author Response
Dear Reviewer and Editor,
Thanks for your comments.
Best regards,
Ahmed R. Alsuwaidi MD, FRCPC, FAAP, FIDSA